



# Measurement Report:concentrations and composition profiles of sugars and amino acids in atmospheric fine particulates: identify local primary sources characteristics

Ren-Guo Zhu[1,3], Hua-Yun Xiao[2,*], Liqin Cheng[1,3], Huixiao Zhu[1,3], Hongwei Xiao[1], Yunyun Gong[1]

[1]Jiangxi Province Key Laboratory of the Causes and Control of Atmospheric Pollution, East China University of Technology, Nanchang 330013, China.
[2]School of Environmental Science and Engineering, Shanghai Jiao Tong University, Shanghai 200240,
China.
[3]School of School of Water Resources and Environmental Engineering, East China University of Technology, Nanchang 330013, China.

*Corresponding author:* Hua-Yun Xiao (Xiaohuayun@sjtu.edu.cn)

**Abstract.** Sugars and amino acids are major classes of organic components in atmospheric fine particles
and play important roles in atmospheric processes. However, the identification of their sources in different regions is less explored. To characterize local primary sources (biomass burning, plant and soil sources) and evaluate their contributions to the total sugar compounds and AAs pool in different regions, fine particulate matter samples were collected from the urban, rural and forest locations. The concentrations and compositions of sugar compounds (anhydrosugars, primary sugars and sugar
alcohols), free amino acids (FAAs) and combined amino acids (CAAs) were analyzed. Overall, the distribution pattern of sugar compounds and CAAs in PM$_{2.5}$ was generally similar among the urban, rural and forest locations. Moreover, the average contribution of sugar compounds and CAAs reflecting BB, plant and soil sources to the total sugar and CAA pool were consistent in all sampling locations. These suggest BB, plant and soil sources all have important contributions to aerosol sugars and CAAs in three
locations. In the urban area, the concentrations of anhydrosugars showed a positive correlation with combined Gly concentrations, but no correlation was found between these two compounds in the rural and forest areas, indicating that the urban area is mainly affected by local combustion sources, while the rural and forest areas may be more influenced by long-transport BB source. In addition, the average L/M ratio in the urban location (59.9) was much higher than those in the rural (6.9) and forest locations (7.2),
implying BB aerosols collected in the urban location originated from lignite burning while the type of biofuels used in the rural and forest locations is mainly softwood. The concentrations of sugar alcohols in the rural and forest locations were positively correlated with that of CAAs, which are abundant in the topsoil, suggesting that the contribution of local topsoil sources is large in these two locations. In the rural area, the concentrations of primary sugars were positively correlated with that of combined aspartic
acid (a CAA specie abundant in grass, the dominant vegetation in the rural area), while in the forest area, primary sugars had good correlations with combined citrulline, lysine, ornithine, glutamic acid and serine (CAA species abundant in pine, dominant vegetation in the forest area), indicating that combining primary sugars with major CAA species in local dominant vegetations may identify local vegetation





types. Furthermore, the nitrogen isotope of combined Gly and PMF model results demonstrated that the

average contribution of combustion processes to total sugar compounds and CAAs pool in the urban and

rural locations was higher than that in the forest location while primary biogenic sources showed a higher

average contribution in the forest location than those in rural and urban locations. Our findings suggest

that combining specific sugar tracers and chemical profiles CAAs in local emission sources can provide

insight to primary sources characteristics including the types of biofuels burned, the contribution of

topsoil sources and local vegetation types.

## 1 Introduction

Organic aerosols (OAs), are recently highlighted because they account for a significant fraction (20%–

90%) of fine aerosols (Verma et al., 2021; Wu et al., 2018; Xu et al., 2020a). Furthermore, they may

influence the global climate as condensation nuclei (CCN) and ice nuclei (IN) (Ng et al., 2010; Steinfeld,

1998), contribute a large amount of terrestrial organic carbon to the remote oceans (Zhu et al., 2015), and

cause human health problems (Nel, 2005; Fu et al., 2008). Thus, a profound understanding of the

abundance, composition and sources of OAs in fine aerosols is of great significance to reduce the adverse

effects of OAs on the atmospheric environments and human being health (Fan et al., 2020).

As the vital fraction of water-soluble organic compounds (WSOC), sugars and AAs are ubiquitous in the

atmosphere from different geographical locations, including urban (Wang et al., 2019), forests (Mace et

al., 2003), mountains (Barbaro et al., 2020; Fu et al., 2008), marine (Fu et al., 2013; Matsumoto and

Uematsu, 2005) and polar regions (Feltracco et al., 2020; Feltracco et al., 2019). Moreover, chemical

fingerprints of these two major classes of organic components in atmospheric particles have been used

to elucidate relative contribution of different sources to aerosols (Jia et al., 2010; Matos et al., 2016; Yan

et al., 2019). Specifically, levoglucosan and related anhydrosugar isomers (mannosan and galactosan),

produced from pyrolysis of cellulose and hemicellulose at temperatures higher than 300℃ and ranking

as the most abundant compounds in biomass burning emissions, have been recognized as specific

molecular markers for biomass burning (BB) sources (Simoneit, 2002). Although there is still lack of

knowledge about BB and amino acids, the potential correlation between the free glycine (Gly)

concentrations and BB aerosols has been suggested in past studies (Zhu et al., 2020a; Samy et al., 2013;

Violaki and Mihalopoulos, 2010). Besides that, BB can elevate the D/L-Ala ratio of atmospheric particles

(Zangrando et al., 2016; Barbaro et al., 2019).

Primary sugars and sugar alcohols are frequently utilized in tracking and estimating the contribution of

soil and associated biogenic aerosols, including fungi, viruses, bacteria, pollen, and plant and animal

debris (Liang et al., 2016). For instance, plant-debris-derived aerosol organic carbon was estimated based

on the relationship of glucose and plant debris (Fan et al., 2020). Sucrose have been widely employed to

indicate airborne pollen grains (Wang et al., 2021; Fu et al., 2012; Yan et al., 2019). The sugar alcohols

arabitol and mannitol, are found to be produced by fungi, lichens, soil biota, and algae to overcome

environmental stress (Medeiros et al., 2006). These sugar alcohols have been recently proposed as

molecular tracers for fungal spores (Elbert et al., 2007). In addition, algae and higher plants are also

proved to be an important source of mannitol (Feltracco et al., 2020). Similarly, the level of combined

amino acids (CAAs) in aerosols has been used as marker for primary biological aerosol particles (PABPs)



and its composition can distinguish different types PABPs based on chemical composition of CAAs in corresponding biological sources (Axelrod et al., 2021; Abe et al., 2016).

Obviously, sugars and AAs in aerosols may share similar source to some extent. However, the concentrations, compositions and source apportionment of sugars and AAs in aerosols have been studied separately in previous studies. Till now, few studies have focused on sugars and amino acids in aerosols simultaneously, and the concentration and composition profiles of sugar compounds and AAs in different areas impacted by different local primary sources is still poorly understood (Barbaro et al., 2019; Ruiz-

Jimenez et al., 2021). It is expected that coupling source specific sugars with most abundant AA species in local primary sources may provide insight into more source signatures.

In recent years, we investigated the source and possible atmospheric transformation process of amino acids and proteinaceous material in different size-segregated particles collected from the forest, rural, suburban and urban sites under different ambient oxidation levels (Zhu et al., 2020a; Zhu et al., 2020b;

Zhu et al., 2021b; Zhu et al., 2021a; Wen et al., 2022). The nitrogen isotope composition of free glycine ($\delta^{15}N_{F-Gly}$) in potentially emission sources were analysed and $\delta^{15}N_{F-Gly}$ values released by BB sources are significantly higher than those from natural primary sources (Zhu et al., 2020a). In the study of Zhu et al. (2020b), the origin of FAAs and CAAs and their atmospheric transformation processes in $PM_{2.5}$ were explored by analysing the concentrations of FAAs and CAAs, and $\delta^{15}N$ values of free and combined Gly

in $PM_{2.5}$ and main primary emission sources (plant, soil, and aerosol from BB). Combined compound-specific $\delta^{15}N$ patterns of HAAs, degradation index with the variance within trophic HAAs in fine and coarse particles, we demonstrated that AAs in coarse particles have advanced bacterial degradation states compared to fine particles (Zhu et al., 2021b). Correlations between the concentrations of FAAs, CAAs and oxidants under different ambient $NO_2$ and $O_3$ levels were discussed to elucidate the mechanism of

FAAs formation from the protein degradation in relation to ambient $NO_2$ and $O_3$ concentrations (Zhu et al., 2021a). Although primary sources are proved to be important sources of AAs in the atmosphere in our previous study, further information of local primary sources characteristics including the type of biofuels burned, local vegetation types and the contribution of topsoil sources are missing.

In order to obtain a sufficient sample size for analysis of sugar compounds (anhydrosugars, primary

sugars and sugar alcohols), FAAs and CAAs in one sample simultaneously, in this study different filter samples were collected during the same sampling period as our previous study. The concentration and composition of sugars, FAAs and CAAs in $PM_{2.5}$ samples collected from the urban, rural and forest areas in Nanchang, China were analysed. Given the documented roles of specific sugars as tracers for BB and PBAPs sources and the distribution of CAAs in primary sources, the variation pattern of the concentration

and composition of specific CAAs species predominant in local primary source and the corresponding sugar tracers were compared. The correlations of source specific sugars with free amino acids (FAAs) and with major CAA species in local primary sources were evaluated (1) to character the concentration and composition profiles of sugar compounds, FAAs and CAAs in urban, rural and forest areas, (2) to obtain the information of the type of biofuels, local vegetation types and the contribution of topsoil

sources in different sampling areas (3) to quantify the contribution of each identified source to total aerosol sugars and CAAs based on positive matrix factorization (PMF).



## 2 Materials and Methods

### 2.1 Sample Collection

PM$_{2.5}$ samples were collected at three different locations (urban, rural and forest) in Nanchang, China from April 30, 2019 to May 13, 2019 (Figure S1). More details of sampling locations are available in supplement material (Table S1).42 Filter samples were collected daily with a duration of 23.5 h on prebaked quartz fiber filters (450 °C, 10h) using high volume air samplers (KC-1000, Qingdao Laoshan Electronic Instrument company, China) at a flow rate of $1.05 \pm 0.03$ m$^3$ min$^{-1}$. Sampling was performed on the roof of buildings (15 m height) at each site. After the sampling the filter was recovered into a pre-combusted glass jar with a Teflon-lined screw cap, then transported to the laboratory and stored at -20°C before analysis, to prevent loss of semi-volatile/volatile organic compounds from the samples.

### 2.2 Analysis of sugar concentrations

For sugars analysis, samples were prepared using a modified version of (Fu et al., 2008). In brief, a filter aliquot (~160 m$^3$ of air) of each aerosol sample was extracted three times with dichloromethane/methanol (2:1; v/v) under ultrasonication for 10 min. The solvent extract was filtered through quartz wool packed in a Pasteur pipette and then dried with pure nitrogen gas. The dried extracts were reacted with 50 ml of N,O-bis-(trimethylsilyl)trifluoroacetamide (BSTFA) containing 1% trimethylsilyl chloride (Sigma-Aldrich, St Louis, MO, USA) and 10 μl of pyridine (Sigma-Aldrich, St Louis, MO, USA) at 70°C for 3 h. After the reaction, derivatives were diluted by the addition of 140 μl of n-hexane containing the internal standard (C13 n-alkane, 1.43 ng μl$^{-1}$) prior to analysis by gas chromatography-mass spectrometry (GC-MS).

GC-MS analyses were performed on a Thermo Scientific TRACE GC (Thermo Scientific, Bremen, Germany) connected into a Thermo Scientific ISQ QD single quadrupole MS. The GC separation was achieved on a DB-5MS fused silica capillary column (30m × 0.25mm i.d., 0.25 μm film thickness) with the same GC oven temperature program: temperature hold at 90°C for 1 min, increase from 90 to 120°C at a rate of 12°C min$^{-1}$, then further increase from 120 to 285°C at a rate of 6°C min$^{-1}$ with a final isotherm hold at 285°C for 18 min. Helium was used as the carrier gas at a constant flow rate of 1.2 ml min$^{-1}$. The temperatures of GC injector and MS ion source were 280 °C and 250 °C, respectively. The mass spectrometer was operated in the electron impact (EI) mode at 70 eV and scanned from 50 to 650 Da. The individual compounds (TMS derivatives) were identified by comparing mass spectra with those reported in the literature and library data and authentic standards (Medeiros and Simoneit, 2007). Compounds were quantified using total ion current (TIC) peak area of each authentic standard relative to that of internal standard (C13 n-alkane), and converted to compound mass using calibration curves of internal standards. In general, the obtained calibration curves show good linearity (R$^2$>0.95) in the exploited range (Table S2). Field blanks were analyzed by the procedure used for the real samples. The data reported here were corrected for the field blanks. In total sixteen sugar compounds, including three anhydrosugars (levoglucosan, galactosan, mannosan), nine primary sugars (sucrose, glucose, fructose, ribose, trehalose, galactose, turanose, lactulose and maltose), and four sugar alcohols (arabitol, mannitol, pinitol and inositol), were detected in the Nanchang aerosols. The detailed method validation was


provided in supporting information Table S2. Recoveries for sugars were better than 89% as obtained for the standards spiked onto the precombusted blank filters and treated as a real sample. The reproducibility of the analytical procedure was assessed through the relative standard deviation (RSD) of the replicate measurements. The RSD values ranged from 1.7 to 11.6%. The detection limits of sugars correspond to ambient concentrations of 0.2-0.9 ng $\mu L^{-1}$, which corresponds to ambient concentrations of 0.2-1.1 ng m$^{-3}$ under a typical sampling volume of 1320m$^3$.


### 2.2 Analysis of FAAs and CAAs concentrations

The concentrations of FAAs and CAAs were analyzed by gas chromatograph-mass spectrometer (Thermo Scientific, Bremen, Germany). For more details of their concentrations analyses refer to our previous publication (Zhu et al., 2020b).

### 2.3 Analysis of water-soluble ions

Water-soluble ions ($Na^+$, $NH_4^+$, $K^+$, $Mg^{2+}$, $Ca^{2+}$, $Cl^-$, $NO_3^-$ and $SO_4^{2-}$) were measured by ion chromatography (Dionex Aquion (AQ)™, Thermo Scientific™, USA). The detection limit of them were 0.001 μg $L^{-1}$, 1.21 μg $L^{-1}$, 1.77 μg $L^{-1}$, 2.47 μg $L^{-1}$, 0.09 μg $L^{-1}$, 5.1μg $L^{-1}$, 21.6 μg $L^{-1}$ and 11.5 μg $L^{-1}$, respectively. The relative standard deviation of the reproducibility test was less than 5% (Guo et al., 2020;

Wen et al., 2022). The mass concentrations of non-sea salt $K^+$ (nss-$K^+$) and non-sea salt $SO_4^{2-}$ (nss-$SO_4^{2-}$) were calculated based on the measured $Na^+$ was assumed to be derived from sea salt as follows (Ren et al., 2018):

nss-$K^+$= $K^+$ -$Na^+$×0.037
nss-$SO_4^{2-}$= $SO_4^{2-}$-$Na^+$×0.252

### 2.4 Statistical Analyses

Statistical analysis and graphs were mainly was conducted by R (version 4.0.2) and Origin 2018 (OriginLab Corporation, USA). Pearson's correlations were conducted to examine the relationship between target sugar category and TFAA, between target sugar category and TCAA, between specific sugar tracers and individual CAA and between specific sugar tracers and individual FAA. Significant

differences in the sugars concentration among three sampling sites were tested using the one-way analysis of variance (ANOVA) procedure, and compared using the LSD test; differences were considered significant at the level of $p < 0.05$.

### 3 Results and discussion

### 3.1 Sugars in the urban, rural and forest locations

### 3.1.1 Sugars concentrations in PM$_{2.5}$

The temporal variations of total sugars concentrations in PM$_{2.5}$ measured at the urban, rural and forest locations during the sampling campaign were shown in Figure 1. The total sixteen sugars (Σsugars) were quantified in the atmospheric fine particulates. Average Σsugars concentration measured in the urban, rural and forest location was 317±139, 181±72, 275±154 ng m$^{-3}$, respectively. Urban location had



significantly higher average Σsugars concentration than that observed in the rural location ($p < 0.05$), but the differences in average Σsugars concentration between rural and forest locations are not significant ($p > 0.05$) (Figure S2).

Three target sugar categories (anhydrosugars, primary sugars, alcohol-sugar) were compared between the three sampling locations. The average concentrations of anhydrosugars in the urban location ($88 \pm$

$56$ ng m$^{-3}$) was higher than those in the rural ($44 \pm 20$ ng m$^{-3}$) and forest locations ($62 \pm 31$ ng m$^{-3}$) (Figure S2). Similarly, the highest average concentration of levoglucosan, a good tracer of BB (Sheesley et al., 2003), was found in the urban location ($83 \pm 53$ ng m$^{-3}$), followed by that in the rural ($37 \pm 16$ ng m$^{-3}$) and forest locations ($54 \pm 27$ ng m$^{-3}$) (Figure 2), indicating enhanced biomass burning activities in the urban location.

Nine primary sugars (fructose, glucose, xylose, sucrose, and trehalose) were measured in this study. The average concentration of total primary sugars measured in PM$_{2.5}$ was comparable in the urban, rural and forest location, with the averaged value of $120 \pm 69$, $71 \pm 47$ and $120 \pm 82$ ng m$^{-3}$, respectively (Figure 3). It is interesting to note that the average concentration of trehalose was higher in the urban location ($28 \pm 66$ ng m$^{-3}$) than those in the rural ($2 \pm 4$ ng m$^{-3}$) and forest locations ($7 \pm 14$ ng m$^{-3}$) ($p < 0.05$) (Figure

S3). Trehalose has been reported to be the most abundant sugar in soils and be used as a tracer for the resuspension of surface soil and unpaved road dust (Fu et al., 2012; Medeiros et al., 2006). Thus, high levels of trehalose observed in the urban location indicates that soil resuspension and unpaved road dust are important sources in the urban location.

We also determined four sugar alcohols, consisting of mannitol, arabitol, pinitol and inositol. The

concentrations of total sugar alcohols averaged $110 \pm 50$, $66 \pm 28$ and $93 \pm 54$ ng m$^{-3}$ in the urban, rural and forest locations, respectively (Figure 4). Average of the sum of mannitol and arabitol concentrations measured in the three sampling locations ($75$ ng m$^{-3}$) were higher than those reported in megacity Beijing ($21$ ng m$^{-3}$) (Kang et al., 2018) and rural site at Lincang, China ($21.5$ ng m$^{-3}$) (Wang et al., 2021). Since these sugar polyols are reported to be abundant in fungi, trees, branches, and leaves (Wang et al., 2021;

Fu et al., 2008), the high abundance of mannitol and arabitol found in this study can be attributed to large amount of biological origin.

The average Σsugars concentration measured in the three sampling locations were comparable to several previous studies, including those measured at urban Beijing, China (average: $200 \pm 100$ ng m$^{-3}$) (Yan et al., 2019), the urban HongKong, China ($292$ ng m$^{-3}$) (Wan and Yu, 2007) and Mangshan National Forest

Park, China ($325$ ng m$^{-3}$) (Verma et al., 2021), lower than those observed at rural Linchang, China ($638$ ng m$^{-3}$) (Wang et al., 2021), rural Wangdu, China ($487 \pm 567$ ng m$^{-3}$) (Yan et al., 2019), but much higher compared to those measured in a natural forest area in India ($156$ ng m$^{-3}$) (Fu et al., 2010b) and Arctic aerosol ($0.6 \pm 0.5$ ng m$^{-3}$) (Feltracco et al., 2020). In summary, the concentrations of aerosol sugars span a large range in variety regions by different research groups, which may be caused by many factors,

including PM$_{2.5}$ concentration, the fraction of organic carbon in PM$_{2.5}$, vegetation and BB activities, and anthropogenic activities (Jia et al., 2010; Verma et al., 2021). To further elucidate the differences in sugar levels in aerosols collected from different regions, source apportionment is necessary.





### 3.1.2 The composition of sugars in PM$_{2.5}$

The distribution pattern of sugar compounds in PM$_{2.5}$ was generally similar among the urban, rural and
forest locations (Figure 5). The average contribution of anhydrosugars to total measured sugars in the
urban, rural and forest locations was 25.8%, 25.9% and 25.0%, respectively (Figure 5). Levoglucosan
was the most dominant species among anhydrosugars in all sampling locations (Figure S4), accounting
for 24.4%, 22.0% and 21.7% of the total sugars pool in PM$_{2.5}$ in the urban, rural and forest locations,
respectively, indicating biomass burning is common in all locations. However, the average contribution
of levoglucosan to the total measured sugars in all locations were lower than those of the aerosol samples
sampled from other regions in China, such as Mountain Tai ($46.1 \pm 20.4\%$ of total sugars in daytime
versus $39.0 \pm 18.0\%$ in nighttime) and fourteen cities in China (around 90%) (Fu et al., 2008; Wang et
al., 2006). This suggests that besides BB sources, other sources are also important sources for sugar
compounds in PM$_{2.5}$ in this study.

The contribution of primary sugars to total measured sugars measured at urban (40.1%) and forest
locations (41.2%), were slightly higher than that in the rural location (36.9%) (Figure 5). Fructose,
glucose and sucrose were the dominant primary sugar species in the three sampling locations (Figure S5).
It is noted that PM$_{2.5}$ sampled from the urban location had a larger contribution from trehalose (an average
6.5% of the total sugars) compared to those sampled from the rural (0.2%) and forest (0.8%) locations
(Figure 5). This result also supports that PM$_{2.5}$ collected in the urban location may be more impacted by
road dust outflow than those in the rural and forest locations.

The average contribution of sugar alcohols to the total sugars pool in PM$_{2.5}$ was also close in the urban
(34.1%), rural (37.2%) and forest (33.8%) locations (Figure 5). Mannitol was the major sugar alcohol
specie in PM$_{2.5}$ (Figure S6), with an average contribution of 26.7%, 26.1% and 24.4% to the total sugars
pool in the urban, rural and forest locations, respectively. Primary sugars and sugar alcohols are reported
to be generally abundant in several animals, vegetative debris, microorganisms, fungal spores and pollen
(Jaenicke, 2005; Jia et al., 2010). The large proportion of primary sugars and sugar alcohols observed in
PM$_{2.5}$ collected from the three locations, further implying ambient sugars in this study were greatly
influenced by primary biological sources.

### 3.1.3 Temporal variation of ambient sugars

Figure S7 presents the overall temporal variations of three classes of sugar compounds (anhydrosugars,
primary sugars and sugar alcohols) in PM$_{2.5}$ collected at the three sampling locations. The temporal
patterns of biomass burning tracers (anhydrosugars), primary bioaerosols tracers (primary saccharides)
and surface soil and associated biotas tracer (sugar alcohols) are similar in the forest location. In the rural
location, the temporal variation of primary saccharides is similar to that of sugar alcohols, but it is not
consistent with that of anhydrosugars. In contracts, anhydrosugars, primary sugars and sugar alcohols
showed different trends in the urban location, indicating that sugar compounds in PM$_{2.5}$ collected in the
urban location might be more influenced by combustion sources whereas ambient sugars in the forest are
more influenced by primary biogenic sources.



### 3.2 AAs in the urban, rural and forest locations

#### 3.2.1 Concentration of specific CAAs

The temporal variations of source-specific sugars and amino acids concentrations in $PM_{2.5}$ were compared in this study. As exhibited in Figure 1, the temporal variation pattern of ambient total free amino acids (TFAA) was different from that of Σsugar concentrations in all locations. Therefore, the concentration and composition of FAAs will not be compared with sugar compounds in the following sections. In contrast, the temporal variation pattern of Σsugar concentrations was consistent with that of TCAA measured in the three locations during the sampling period (Figure 1). Furthermore, there were similar time variation patterns in the concentrations of anhydrosugars, combined Gly and Phe (Figure 2), indicating anhydrosugars, combined Gly and Phe may be influenced by the identical source. The temporal variation pattern of primary sugars concentrations was similar to those of the Asp and the sum of combined Cit, Lys, Orn, Gly and Ser (Figure 3), implying primary sugars may share the similar source with combined Asp, Cit, Lys, Orn, Gly and Ser. Additionally, the time variation pattern of sugar alcohols concentrations was consistent with that of the sum of Ala, Val, Leu and Ile concentrations (Figure 4), suggesting sugar alcohols and combined Ala, Val, Leu and Ile may be derived from the same sources.

In our previous study, the composition profiles of CAAs in the biomass burning (honeycomb briquette, pine, and straw burning), soil, and plant sources (pine and straw) were investigated. Gly was found to be the predominant CAA in the aerosol from biomass burning sources, hydrophobic species (Ala, Val, Leu, and Ile) were the major CAA species in soil sources, and hydrophilic species (Glu, Lys, and Asp) were the abundant CAA species in plant sources (Zhu et al., 2020b). Therefore, based on the specific CAAs species predominant in each emission source and temporal variations pattern of source-specific sugars and CAA species, ambient CAAs were divided into 4 groups: the first group includes Gly and Phe, which are closely related to the BB source; the second group includes Ala, Val, Leu, and Ile, which are greatly impacted by soil source; the third group including Asp, Cit, Lys, Orn, Glu, Ser, which is close with plant sources, and the other CAAs are classified into the fourth group (Figure 5).

The sum of Phe and Gly concentrations (group 1) in $PM_{2.5}$ averaged $58 \pm 46$ ng m$^{-3}$, $55 \pm 33$ ng m$^{-3}$ and $57 \pm 38$ ng m$^{-3}$ in the urban, rural and forest locations, respectively (Figure 2). Average concentration of the sum of Ala, Val, Leu, and Ile (group 2) in urban, rural and forested mosses were $110 \pm 75$ ng m$^{-3}$, $85 \pm 40$ ng m$^{-3}$ and $79 \pm 45$ ng m$^{-3}$, respectively, and average concentration of the sum of Asp, Cit, Lys, Orn, Glu, Ser (group 3) were $123 \pm 135$ ng m$^{-3}$, $78 \pm 53$ ng m$^{-3}$ and $93 \pm 68$ ng m$^{-3}$, respectively (Figure 3 and Figure 4). By comparing the concentration of CAAs in different groups, it can be found that the average of the sum of CAAs concentrations in group 2 and group 3 is higher than that in group 1. This suggested that similar to sugar compounds, CAAs in $PM_{2.5}$ collected from the three sampling locations were greatly impacted by plant and soil sources in addition to biomass burning source.

#### 3.2.2 Composition profiles of specific CAAs at the three sampling sites

Combined Gly and Phe together accounted for an average of 19.7%, 23.9% and 20.4%, respectively, of the TCAAs pool in $PM_{2.5}$ collected from the urban, rural and forest locations. It is interesting to note that the percentage of Gly and Phe (abundant in BB source) in CAAs pool was generally similar to the



percentage of the anhydrosugars in total sugar compounds in PM$_{2.5}$. Ala, Val, Leu, and Ile (abundant in soil source) together contributed an average of 35.0%, 33.3% and 31.1%, respectively, to the TCAAs

pool in the urban, rural and forest locations, which was also similar to the contribution of sugar alcohols to the total sugar compounds in the three locations. Furthermore, the average contribution of the combined CAA species in groups 3 and 4 to the TCAA pool was similar to the percentage of primary sugars to total sugar compounds in the three sampling locations (Figure 5). It is likely that composition profiles of sugar compounds and CAAs in PM$_{2.5}$ are mainly controlled by their primary sources.

### 3.3 Comparative behaviour of sugars and AAs in aerosols

#### 3.3.1 Implications of formation pathways and atmospheric processes of ambient sugars and AAs

Pearson's correlation coefficients were performed based on sugar and AAs concentrations in PM$_{2.5}$ samples aggregated by sampling sites, and the results are shown in the Figure 6. It is interesting to note that no correlation was observed between total free amino acids (TFAA) and the other individual

measured sugar and total sugars in the three sampling locations (Figure 6a, yellow shadow). This could be explained by the formation pathways and atmospheric processes of aerosol FAAs and sugars. The fact that secondary processes involved in the photochemical oxidation of atmospheric proteinaceous matter is an important formation pathway of FAAs in the atmosphere has been corroborated by many studies (Wang et al., 2019; Xu et al., 2020b). Laboratory simulation experiments observed that the oxidation of

proteins and peptides with hydroxyl radicals (·OH) and O$_3$ can release of FAAs upon in aqueous phase (Liu et al., 2017b; Mcgregor and Anastasio, 2001). Song et al. (2017) also demonstrated that O$_3$ can also induce more proteinaceous matter to release FAAs in the atmosphere. On the other hand, sugar compounds are suggested as atmospheric primary organic aerosol, which directly emitted from sources such as plant material, soil dust, microorganisms and BB, and then aerosolized by active release from

biological organisms, wind suspension, droplet impaction or human activities such as agricultural activities (Fröhlich-Nowoisky et al., 2016; Joung et al., 2017). Moreover, compared to FAAs, sugars are not derived from secondary photochemical processes in the atmosphere.

To further investigate the correlation between ambient Σsugar concentrations and FAAs, Pearson's correlation coefficients were calculated based on total sugar concentrations and individual FAA

concentrations in PM$_{2.5}$ aggregated by sampling locations, which are exhibited in Figure S8. Generally, no correlation was recorded at the three sampling sits (Figure S8). Only Gln, Cit, Phe and Thr were weakly correlated with the concentration of total sugar concentrations in the forest location, and Glu exhibited a weak correlation with total sugar compounds in the rural location. These individual FAAs are known to be relatively high reactive with short half-lives in the atmosphere (Mcgregor and Anastasio,

2001). It is likely that these individual reactive FAAs are directly emitted from biological sources rather than secondary processes. Accordingly, we infer that there are differences in the formation pathways and atmospheric processes of sugars and FAAs in fine particles.

On the contrary, good correlations between total combined amino acids (TCAA) and total sugars (r=0.56~0.57, $p < 0.05$) were obtained at three sampling sites in this study (Figure 6a). Although the

relationship between CAAs and sugars in the atmosphere has not been directly analysed, their sources have been explored in previous studies. For instance, Feltracco et al. (2019) investigated the possible



sources of AAs in Artic aerosol and proved FAAs and CAAs in aerosols might have different sources. They suggest that similar to aerosol sugars, ambient CAA derive from PBAPs, including viruses, algae, fungi, bacteria, protozoa, spores and pollen, fragments of plants and insects, and epithelial cells of

animals and humans. Previous studies also demonstrated that unlike FAAs greatly influenced by atmospheric processes, composition profiles of aerosol CAAs are highly depended on the origin of CAAs and it has been used to trace emission sources including primary biological sources and BB sources (Abe et al., 2016; Zhu et al., 2020b). Thus, good correlations between TCAA and total sugars in aerosols found in this study further corroborate that they may have common sources and are less impacted by

atmospheric photochemical processes.

### 3.2.2 Evaluate the contribution of primary emission sources in different locations.

Sugars in the atmosphere have been demonstrated to be derived from numerous sources. Three target sugar categories (anhydrosugars, primary sugars and sugar alcohols) can be used as tracers for specific source information (Simoneit, 2002; Simoneit et al., 2004). Anhydrosugars (including levoglucosan,

mannosan and galactosan), generated by the thermal decomposition of cellulose and hemicellulose, are regarded as the specific markers for BB in the atmosphere (Simoneit et al., 2004; Zangrando et al., 2016). In this study, TCAA was found to be highly correlated with anhydrosugars at the urban and rural locations whereas no correlation between TCAA and anhydrosugars was observed in the forest location (Figure 6a, red shadow), indicating the urban and rural areas were more influenced by BB sources compared to

the forest area. These conclusions were also corroborated by the $\delta^{15}N$ signatures of combined Gly ($\delta^{15}N_{C-Gly}$) in $PM_{2.5}$ measured at these three sites. Our previous study found that more positive $\delta^{15}N_{C-Gly}$ values were observed in $PM_{2.5}$ at the urban (average = +8.2 ± 5.4‰) and the rural (average = +5.9 ± 3.2‰) site than the forest site (average = +3.3 ± 6.2‰) (Zhu et al., 2020b) (Figure 6b). According to the $\delta^{15}N$ inventories of emission sources of atmospheric combined Gly, the urban location was highly influenced

by combustion sources, followed by the rural location, while the forest location was more affected by natural sources.

Primary sugars are reported to be derived from microorganisms, plants, animals, lichens, and bacteria and have been considered to be good tracers for PBAPs (Fu et al., 2013; Zangrando et al., 2016). Particularly, sucrose and fructose are predominantly produced from plant fragments (e.g., bark, pollen

grain, flowers buds, fruit and leaves) and soil dust (Fu et al., 2012; Xu et al., 2020a). In this study, TCAA only exhibited positive correlation with primary sugars (fructose and sucrose) in the forest location (Figure 6a, purple shadow), pointing to aerosols in the forest area were mainly influenced by intensive biological activities and vegetation growth during sampling period. Previous studies mentioned that increased biological activities and active vegetation such as developing flower buds in the spring can

emit abundant primary sugars into the air (Jia et al., 2010). Hence, the sampling period (spring) of this study and intensive biological sources in the forest location may explain the positive correlation between TCAA and primary sugars in the forest area.

Sugar alcohols primarily come from fungal spores, bacteria, plants, algae and detritus of mature leaves (Bauer et al., 2008; Feltracco et al., 2020). Xu et al. (2020a) suggest that in spring mannitol and arabitol

may be closely related to the activities of the terrestrial biosphere. When comparing the correlation results





of sugars classes between sites, it is important to note that the correlation of primary sugars with sugar alcohols varied degrees at different sampling sites (Figure 6a, green shadow). More species of primary sugars and sugar alcohols in $PM_{2.5}$ positively correlated at the rural and forest sites ($p < 0.05$) compared to the urban site. Moreover, the stronger correlation between primary sugars with sugar alcohols were

observed for $PM_{2.5}$ samples collected from the rural and forest sites than that at the urban site. This may imply that the impact from local biogenic sources at the rural and forest sites was greater than that of the urban site, which is also supported by the temporal patterns of anhydrosugars, primary sugars and alcohol-sugar at the three sites (Figure S7). We can thus conclude that the influence of primary biogenic and BB sources on aerosols in different locations can be evaluated by the correlations between TCAA

and source specific sugar tracers (anhydrosugars, primary sugars and sugar alcohols).

### 3.2.3 Identify burning tissues, local vegetation classes and soil-related biological sources

A Pearson's correlation test was also calculated for the dataset containing ambient concentrations of sugar compounds and individual CAAs in $PM_{2.5}$ collected at three sites, which are exhibited in Figure 7. In the urban location, the concentrations of anhydrosugars showed a positive correlation with that of combined

Gly, but no correlation was found between these two compounds in the other two locations. As we discussed above, combined Gly is major CAA specie in the honeycomb briquette, pine, and straw burning aerosols. Therefore, anhydrosugars and combined Gly both are closely related to biomass burning sources. A possible explanation for the positive correlation found between these two BB tracers is that the urban region is mainly affected by local combustion sources, while rural and forest regions may be

more influenced by BB source from long-transport.

Significant positive correlations of anhydrosugars with Phe were observed simultaneously in the urban (r=0.68, $p<0.05$), rural (r=0.65, $p<0.05$) and forest (r=0.57, $p<0.05$) locations, serving as statistical support for a common source for the combined Phe and anhydrosugars in aerosols (Figure 7). Recent studies indicate that the amino acid Phe is not only used as vital constituent of proteins but also a critical

metabolic node that plays an essential role in lignin biosynthesis in vascular plants (Bonawitz and Chapple, 2010; Wallace and Fry, 1994). It is estimated that nearly 30–45% of photosynthetically fixed carbon is channelled through Phe for the biosynthesis of lignin during wood formation (Pascual et al., 2016). In addition, the vascular plant cell wall is a heterogeneous natural nanocomposite of cellulose, hemicelluloses and lignin (Agarwal, 2006; Jin et al., 2017). The lignin, the second most abundant

polymer in plants, deposited in the secondary wall in which cross-linked to the structural polysugars, such as hemicelluloses and cellulose found there (Bonawitz and Chapple, 2010; Wallace and Fry, 1994). During the combustion processes of plant cell wall, the thermal breakdown of lignin takes place along with cellulose/hemicellulose. Therefore, the combustion of plant cell wall may explain the observed concentration coupling between combined Phe and anhydrosugars.

In this study, primary sugars were found to be highly correlated with combined Asp in $PM_{2.5}$ samples collected in the rural location, but primary sugars were correlated with combine Glu, Lys, Orn, Cit and Ser in the forest location (Figure 7, blue shadow), which is consistent with the temporal variation pattern of primary sugars and these CAA species (Figure 3). This may be attributed to the difference vegetation types in these two regions. The rural site is a typical agricultural area where grasses and crops are the





main vegetation classes. Asp is reported to be the most abundant free and combined amino acids in Gramineae (Barnett and Naylor, 1966) (Burkholder et al., 1959; Wilson and Tilley, 1965). On the other hands, conifers dominate ecosystems at the forest site. In particular, hydrophilic species (Glu, Lys, Orn, Cit and Ser) have been shown to be the predominant CAA species in pine needles (Zhu et al., 2020b). Hence, the correlation between primary sugars (primary bioaerosols tracers) and specific CAAs may be

used to track the predominant local vegetation types.

Consistent correlations were observed between sugar alcohols and combined Ala, Val and Leu at the rural and forest sites (Figure 7, green shadow), pointing to the significant contribution of biological sources associated with surface soil to aerosols at these two sites. As above mentioned, sugar alcohols indicate soil and associated biota contribute to ambient aerosols through resuspension by wind erosion

or agricultural activities (Fu et al., 2013; Fu et al., 2010a). Similarly, Friedel and Scheller (2002) showed that Ala, Val and Leu were the main hydrolysable amino acid species in soil organic matter and soil microbial biomass. Grass land environments are characterized by higher content of Ala, Asp and Leu (Moura et al., 2013). Our previous studies also demonstrated that hydrophobic Ala, Val, Leu, and Ile were the major CAA species in local crop soil (Zhu et al., 2020b). On the contrary, primary sugars and

sugar alcohols exerted no correlation with individual CAA at the urban site except for Phe, further supporting that aerosol at urban location are less influenced by biogenic source than combustion sources. Thus, it can be concluded that the correlations between individual CAAs and target category sugars can provide insight into various source characteristics such as plant tissues used for combustion, local vegetation classes (based on the correlation of primary sugars with specific CAAs dominant in local

plants) and soil-related biological sources vs combustion sources (based on the correlation of sugar alcohols with specific CAAs dominant in soil source).

### 3.3 Source identified by PMF

To investigate the source apportionment of sugars and TCAA in $PM_{2.5}$, positive matrix factorization (PMF) software version 5.0 (Environmental Protection Agency, USA) was used. The PMF analysis was

performed for twenty-five tracer compounds, including anhydrosugars, primary sugars, sugar alcohols, water-soluble ions, TFAA and TCAA. As discussed in section 3.2.1, the sources and atmospheric photochemical processes of individual FAAs are different from sugar compounds and CAAs, thus individual FAAs were also excluded from the PMF. We adopted 15% of sugars and combined CAA concentration values as their measurement uncertainties (Fu et al., 2012; Xu et al., 2020b). The

measurement uncertainties (10%) were used for the error estimates of the measured values of water-soluble ions. To maximize the quantity of data, half of the LOD was used for the values below the detection limit and 5/6 of the LOD was used for the corresponding error estimate (Wen et al., 2022).

The number of factors can be determined by the change of Q (robust) and Q (true). In this study, based on the possible sources of sugars and TCAA, four to six factor solutions were run in PMF model. In

moving from four to six factors, the decrease of Q/Qexpected is illustrated from 4.7 to 3.2 in the urban location, from 3.8 to 2.9 in the rural location, and from 3.5 to 2.5 in the forest location. When the changes in Q values are small with the increase of factors, it can suggest that there may be too many factors being fit, indicating that four factors could be the appropriate factor number as to the probable origin of total





sugars and TCAA at three sampling sites (Liu et al., 2017a). These four factor solutions were preferred

based on minimum robust and true Q values (goodness of fit parameters) of the base runs, which observed 753.9 and 846.3, 669.6 and 684.8, 641.3 and 683.5, at the urban, rural and forest sites respectively. In each bootstrap run, the concentrations of tracers were close to those of base-run results.

As shown in Figure 8, factor 1 is characterized by the high contribution of levoglucosan and nss-$K^+$. Levoglucosan has been tested and confirmed as an excellent tracer for the biomass burning, which

produce from the pyrolytic decomposition of cellulose and hemicellulose, or thermal stripping of sugar polyols during the combustion processes (Jia et al., 2010; Kang et al., 2018; Simoneit, 2002). nss-$K^+$ is also reported to be a tracer of biomass burning (Kunwar and Kawamura, 2014). Besides that, it was noted that CAAs and TFAA also had a relatively important proportion in factor 1, particularly for combined Glu and Ser in the urban sites. Previous studies showed that the concentration of fungal material in $PM_{2.5}$

elevated during BB events and viable fungal spores have been identified in the smoke of biomass burning (Mims and Mims, 2004; Yang et al., 2012). Wei et al. (2019) attributed this result to turbulent air during the burning event can transport bacterial, fungal cells, various metabolism microbes and human or plant pathogens into atmosphere. Moreover, protein and AAs are widely distributed in these organisms (Dauwe et al., 1999). In urban site, anthropogenic origin and combustion-related elemental carbon are easily

coated with a layer of peptides and protein (Filippo et al., 2014). Matsumoto et al. (2021) also suggest that combined Glu in fine particle was derived from biomass burning particles both from long transport and local sources. Thus, factor 1 can serve as an indicator of the influence of BB.

Factor 2 is dominated by high loading of levoglucosan, $Ca^{2+}$ and $Mg^{2+}$. In addition, mannitol and some saccharides contributed a high proportion to factor 2, especially in the urban and forest sites. Previous

studies reported that mannitol and these primary saccharides were close related to biologically active surface soils and unpaved road dust (Wang et al., 2021; Rogge et al., 2007). Similarly, $Mg^{2+}$ and $Ca^{2+}$ are mainly derived from resuspended dust/soil in the ambient air (Barbaro et al., 2019; Rathnayake et al., 2016). Recently studies demonstrate that levoglucosan is also present in lignite smoke particle matter (Rybicki et al., 2020a). Moreover, pyrolysis experiments confirm that lignite produce levoglucosan

concentrations fully comparable to that of hardwoods and softwoods (Fabbri et al., 2009; Sheesley et al., 2003). Burning of lignite has been proved to be an additional input of levoglucosan to the atmosphere in regions where brown coal is utilized. Moreover, compared to factor1, factor2 shows high loadings of $Cl^-$. $Cl^-$ has been reported to be a major chemical composition of coal combustion. It is reasonable to assume that burning substrates associated with factor 1 and factor 2 might be different. In order to confirm this

idea, we analyzed the concentration ratios of levoglucosan to mannosan (L/M) in the three sampling locations (Figure 10). L/M ratios have been employed to identify specific burning substrates in numerous studies (Fabbri et al., 2009; Fan et al., 2020; Zhu et al., 2015). For instance, high L/M ratios were reported for smoke particles from the burning of crop residues (~40 to 55.7), while those from softwood (3 to 10) and hardwood (15-33) combustion were characterized by low L/M ratios (Engling et al., 2014; Engling

et al., 2009; Sheesley et al., 2003; Wang et al., 2021). However, even higher L/M ratios (>57) were observed at the urban sites (Figure 10). Rybicki et al. (2020b) showed that the levoglucosan/mannosan (L/M) ratios in lignites smoke (ranged from 31 to 189) are generally higher than those found for other fuel types because cellulose is more resistant to diagenetic (bio)degradation than hemicelluloses. These





results may imply that factor 2 represents a mixture source of lignite combustion and resuspended
dust/soil.

factor 3 is characterized by the high contribution of $NH_4^+$, nss-$SO_4^{2-}$, $NO_3^-$, and TFAA. Both $NH_4^+$ and
nss-$SO_4^{2-}$ are reported to be the secondary particulate components that are converted from gas pollutants
(Dai et al., 2013; Guo et al., 2020). Additionally, secondarily process is an important pathway of the
formation of atmospheric FAAs (Xu et al., 2020b). Thus, Factor 3 can be identified as secondary
processes.

Factor 4 was loaded with CAAs and primary sugars. CAAs are the basic component of primary biological
aerosol particles, including viruses, algae, fungi, bacteria, protozoa, spores and pollen, debris of plants
and insects, and epithelial cells of animals and humans (Abe et al., 2016; Matos et al., 2016). Primary
sugars are also reported to be generally more abundant in vegetation detritus, airborne pollen and
microorganisms (Jia et al., 2010; Xu et al., 2020a). Thus, this factor represents the primary biogenic
source.

Overall, the average contributions of each factor to total sugars were estimated by PMF analyses (Figure
9). The two factors associated with combustion processes (factor 1 and factor 2) contributed higher to
total sugars in $PM_{2.5}$ at the urban and rural sites than the forest site. This result is consistent with $\delta^{15}N$
signatures of combined Gly in $PM_{2.5}$ and the correlation between anhydrosugars and TCAA in PM2.5
(Figure 6). More specifically, BB was found to account for 22.6%, 45.5% and 30.55% of total sugars at
urban, rural and forest sites. On the contrary, coal combustion and road dust sources contribute a greater
fraction to total sugars at urban site (53.9 %), compared with 18.6% at rural and 8.3% at forest site. This
is supported by the L/M ratios. Higher L/M ratios were found in the urban location whereas lower ratios
were observed in the rural and forest locations (Figure 10). In this study, the ratios of L/M in the rural
and forest locations ranged from 5.1 to 8.4 (average: 6.9) and from 4.3 to 9.8 (average: 7.2), respectively,
indicating major contribution from softwood burning at these two regions. The extremely high L/M ratios
found in the urban location (ranged: 7.9 to 359.1; average: 59.9). These ratios were even higher than the
values in the smoke particles released by the burning of crop residuals reported in the previous literature
but fell in the range of lignite combustion, implying the urban site may be influenced by lignite
combustion besides biomass burning. Further support for this result was provided by the fact that a great
quantity of lignite was used at the urban site (averaged 3803646 tons per year, the data are from the
statistical yearbook of Nanchang city, http://tjj.nc.gov.cn//zbft/front/tjjnjnew/2020/mobile/index.html).
As mentioned above, factor2 is a mixture source of lignite combustion and resuspended dust/soil. Greater
impact of resuspended dust/soil on total sugars in $PM_{2.5}$ at the urban site can be corroborated by higher
concentration (Figure S3) and abundance of trehalose (Figure 5) measured in $PM_{2.5}$ in the urban location
compared to those found in the rural and forest locations.

In addition, primary biogenic sources (factor 4) had a larger contribution in the forest location (53.7%)
than those in the rural (8.4%) and urban (8.3%) locations. The urban (15.2%), rural (27.5%), and forest
(7.5%) sites exhibited relatively low secondary processes contributions (factor 3), paralleling the
correlation result between the total sugars and TCAA (Figure 6a), which confirming that ambient sugars
and CAAs are mainly derived from primary sources.



We also analyzed the proportional contributions to the TCAA in PM$_{2.5}$ in the three sampling locations (Figure 9). The primary difference among the three locations lies in the contributions from BB and biogenic sources. Fractional contributions of aerosol TCAA from BB in the urban (41.9%) and rural (70.3%) locations were significantly higher than that in the forest location (12.1%), but TCAA from biogenic source in the urban (34.4%) and rural (15.2%) locations were significantly lower than that in the forest (42.4%) location. Similar to total sugars in PM$_{2.5}$, there are relatively low secondary processes contributions for TCAAs in the three sampling locations (lower than 20.2%).

**4 Conclusions**

Determining the presence and signatures of source specific sugars and AAs in aerosol samples can be of great value to estimate the contribution of primary biogenic and BB sources to aerosols at different sampling locations. Anhydrosugar, a confirming BB tracer, highly correlated with combined Phe suggests the burning of plant cell wall. The correlations between primary sugars (primary bioaerosol tracer) and specific CAAs dominant in local plant classes can help to identify potential local plant classes. Combined sugar alcohols (soil and associated biota tracer) with specific CAAs species dominant in soil source can help to distinguish soil-related biological sources contribution. Previous work on aerosols focused on using only sugars or AAs biomarker alone to trace the aerosol sources. To our knowledge, this is the first report combining source specific sugar tracers with amino acids species dominant in primary emission sources to trace primary sources signatures at different locations. Moreover, this paper clearly demonstrates combining these two molecular biomarkers in atmospheric particles may provide more source characteristics than using a single biomarker alone. Future work should be aimed at comprehensively characterizing physical, chemical, and biological properties of sugars and amino acids in aerosols, assessing possible interaction mechanisms between them, and extensively addressing the distribution of these two compound classes in potential aerosol sources.

*Data availability.* The data for lignite consumption at the urban site were obtained from the statistical yearbook of Nanchang city (http://tjj.nc.gov.cn/zbft/front/tjjnjnew/2020/mobile/index.html) (Nanchang Statistics Bureau, 2020). Raw data sets (Zhu, 2021) used in this manuscript were available at https://figshare.com/articles/dataset/Concentration_of_saccharides_in_PM2_5_xlsx/17158661.

*Supplement.* The supplement related to this article is available online.

*Author contributions.* This research was designed RGZ and HYX. Laboratory measurements were performed by HXZ, LQC and YYG. The paper was prepared by RGZ, HYX and HWX.

*Competing interests.* The authors declare that they have no conflict of interest.

*Financial support.* This work was supported by the National Natural Science Foundation of China (Grant No.41425014 and 41463007).





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





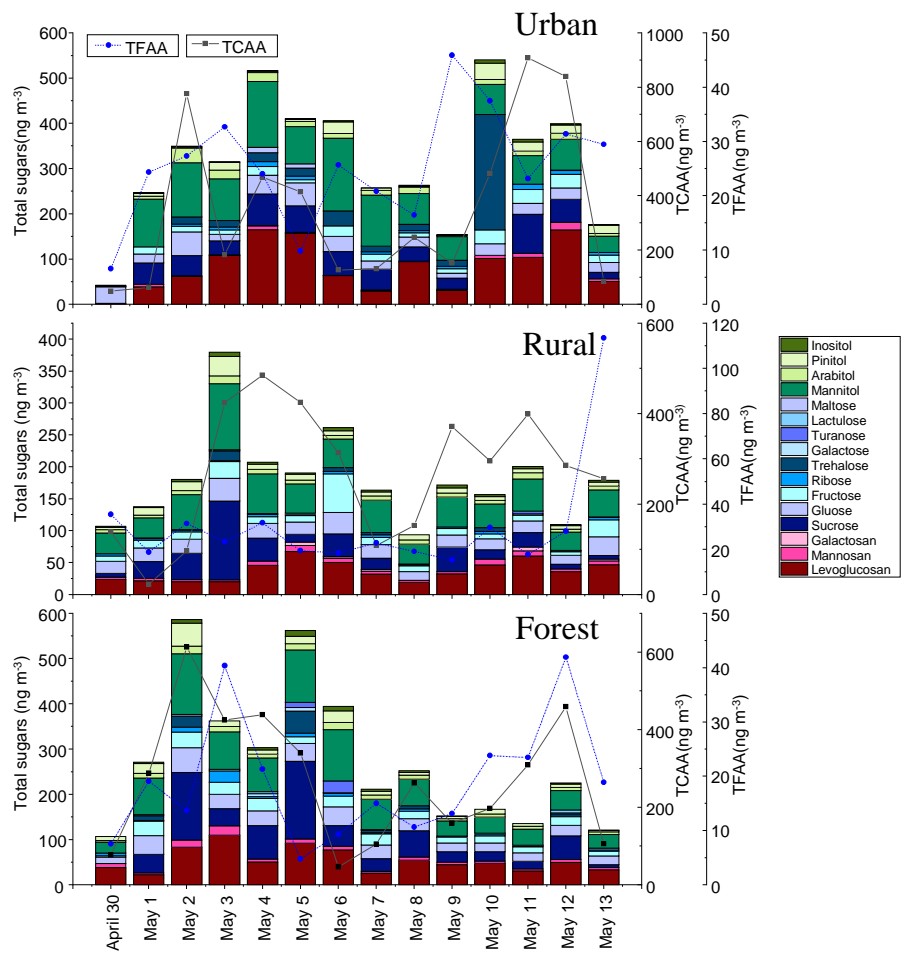

**Figure 1. Temporal variations in the concentrations of total sugars and total combined amino acids measured in PM2.5 samples collected at the urban, rural and forest sites in Nanchang, China.**








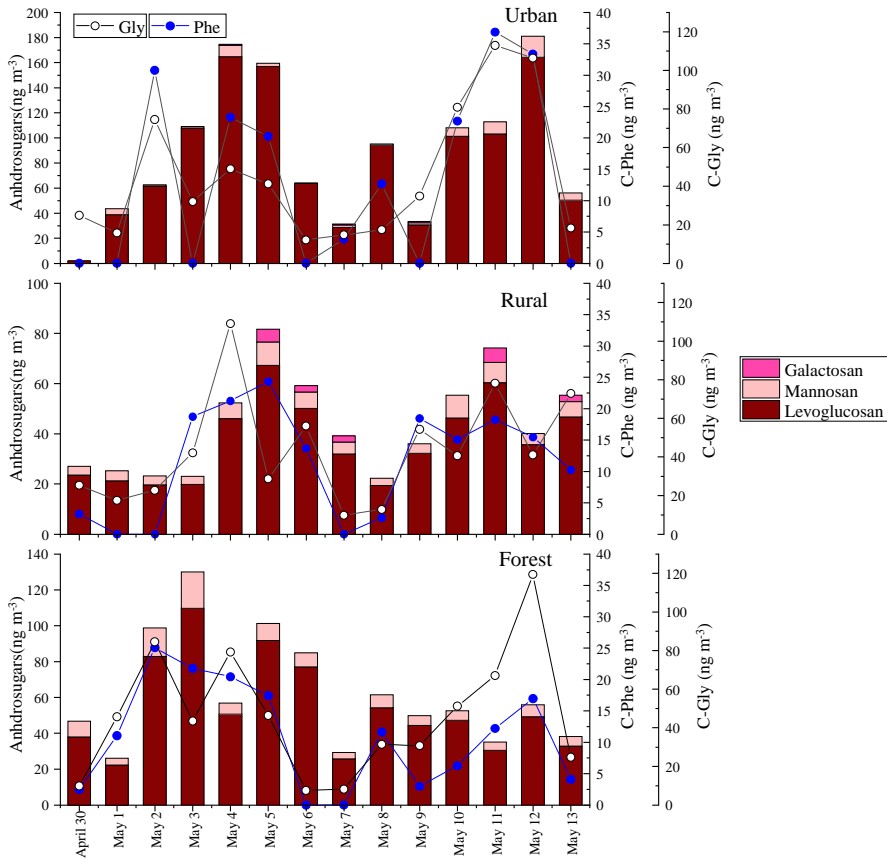

**Figure 2. Concentrations of anhydrosugars, combined Phe and Gly in urban, rural and forest. The concentrations of anhydrosugars and CAAs for each sample were normalized for the total volume of air sampled.**









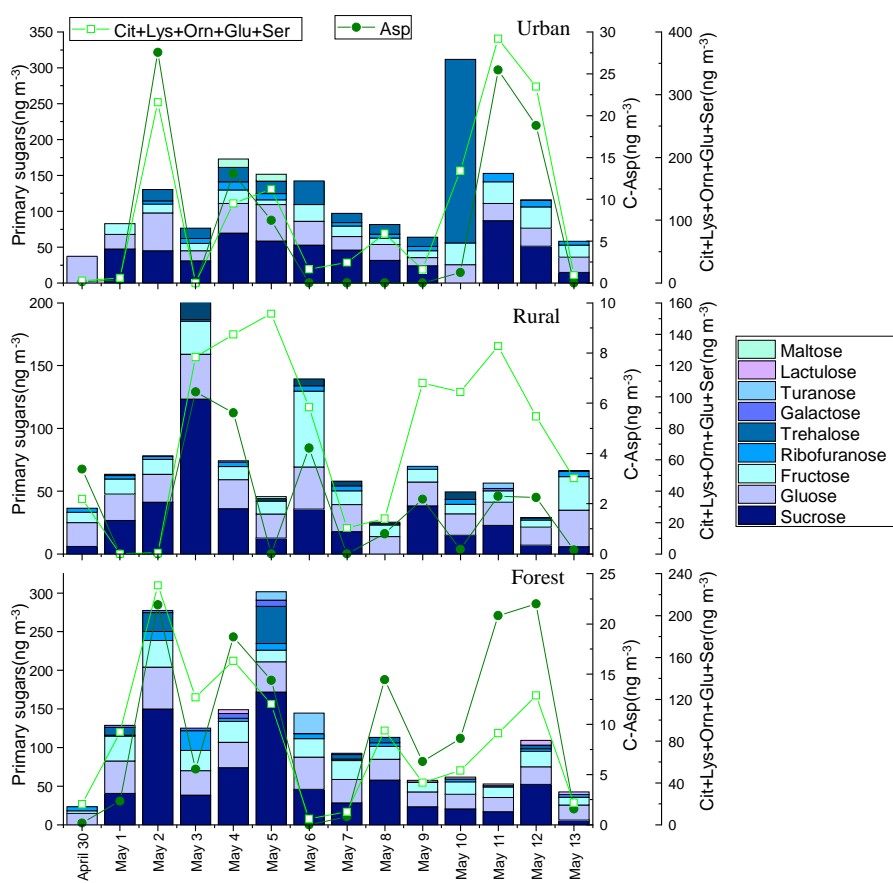

**Figure 3. Concentrations of primary saccharides, combined Asp and the sum of Cit, Lys, Orn, Glu and Ser in urban, rural and forest. The concentrations of primary saccharides and CAAs for each sample were**

**normalized for the total volume of air sampled.**







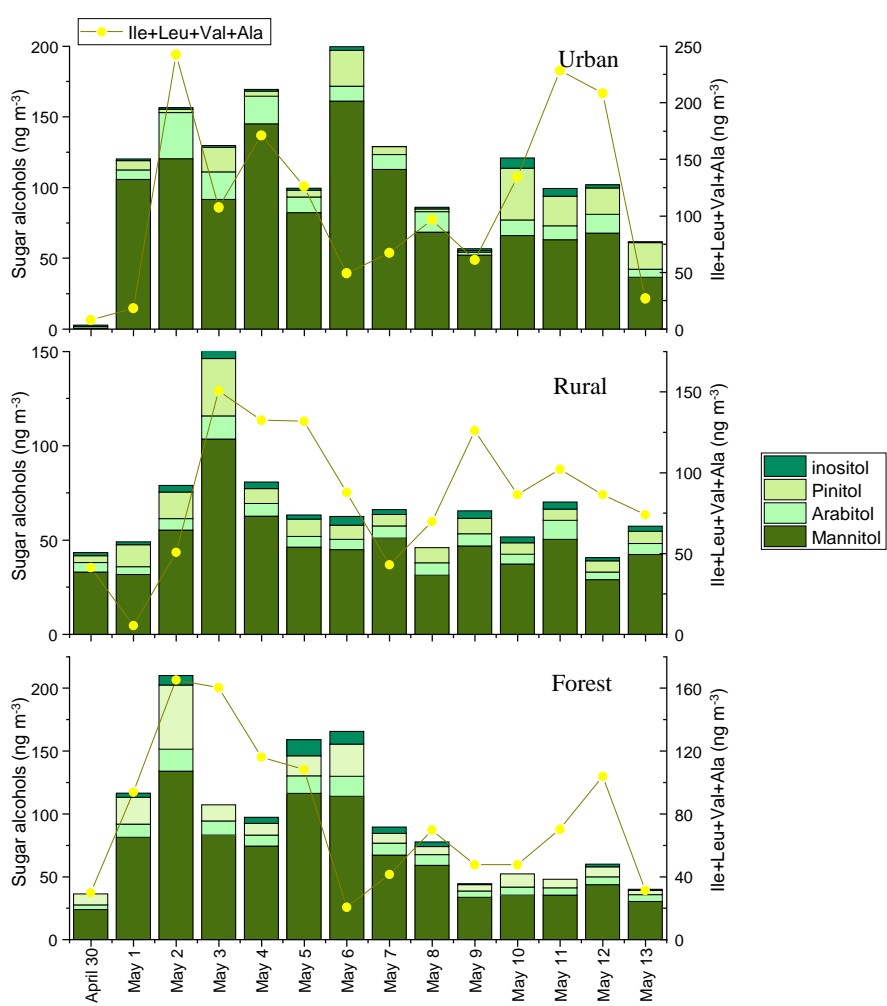

**Figure 4. Concentrations of sugar alcohols the sum of Ile, Leu, Val and Ala in urban, rural and forest. The concentrations of sugar alcohols for each sample were normalized for the total volume of air sampled.**





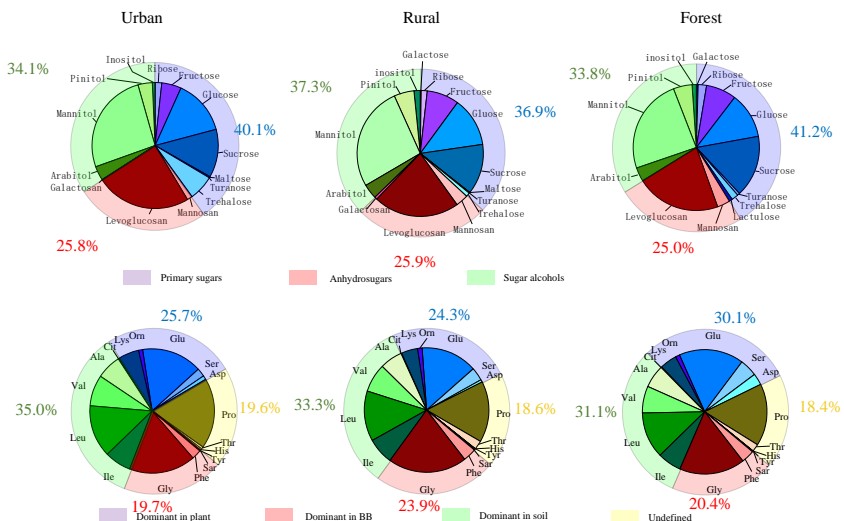


**Figure 5. Illustrations of the average contributions of individual species to total sugars and those of each CAA to total combined amino acids in urban, rural and forest sites.**










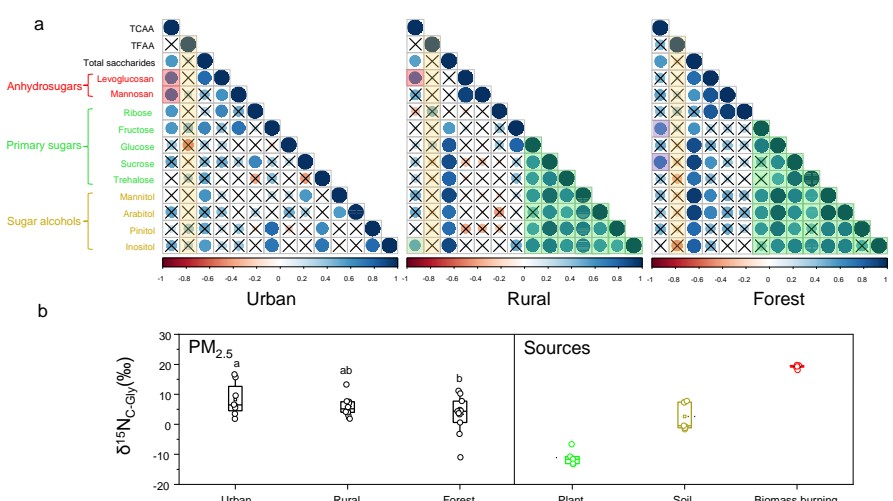

**Figure 6. (a) Pearson correlations between TFAA and sugar species, and between TCAA and sugar species in PM$_{2.5}$ collected in urban, rural and forest sites. The cross indicates a $p$-value higher than 0.05. The ball indicates a $p$-value less than 0.05. The larger a ball is, the more significant the correlation is; (b) The range of δ$^{15}$N for combined Gly in PM$_{2.5}$ and major emission sources. The data of combined Gly was cited from Zhu et al. (2020). Open circles represent the isotopic composition data of combined Gly.**










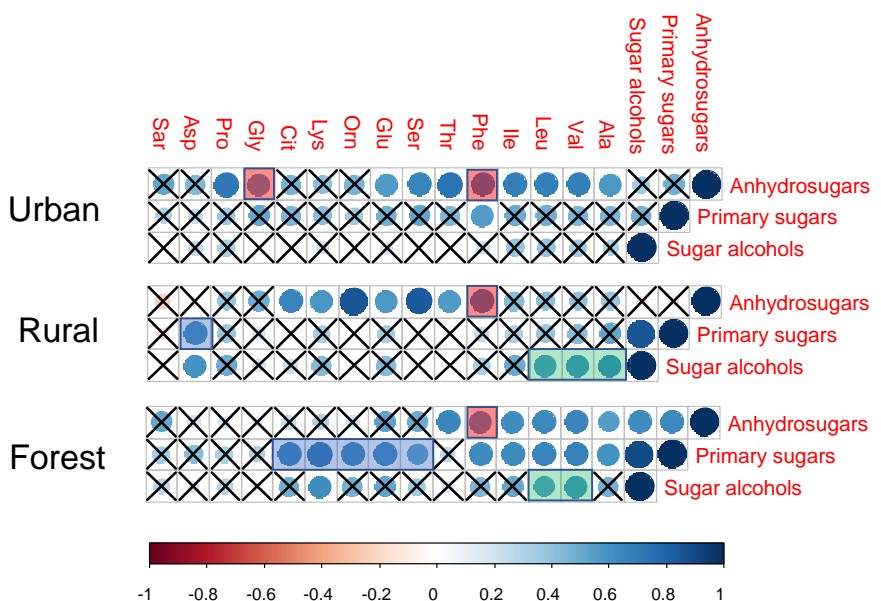

**Figure 7. Pearson correlations between three target sugars (anhydrosugars, primary sugars and sugar alcohos) and individual combined amino acids. The cross indicates a *p*-value higher than 0.05. The ball indicates a p-value less than 0.05. The larger a ball is, the more significant the correlation is.**





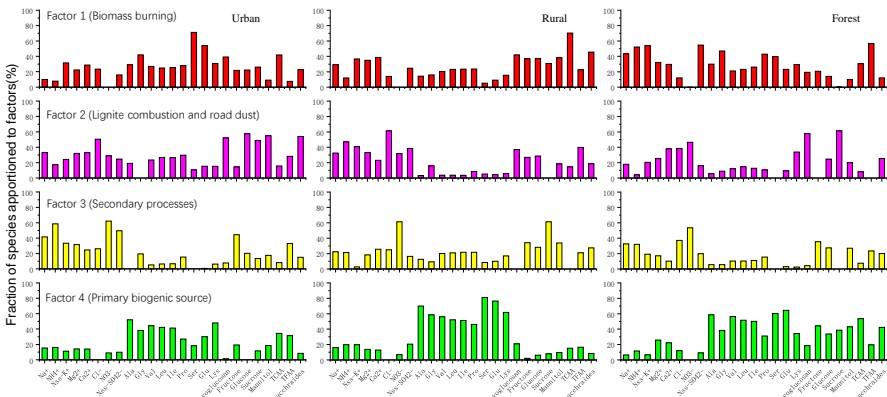

**Figure 8. Profiles of four sources obtained with the PMF. Percentage contribution of each factor to the concentration of each species.**





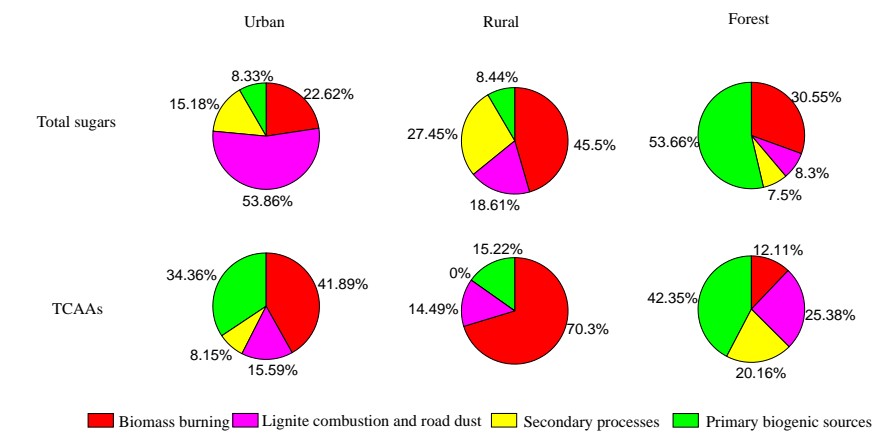


**Figure 9. Illustrations of the percentage of factor contributions to the total sugars and CAAs in PM$_{2.5}$ obtained by PMF.**











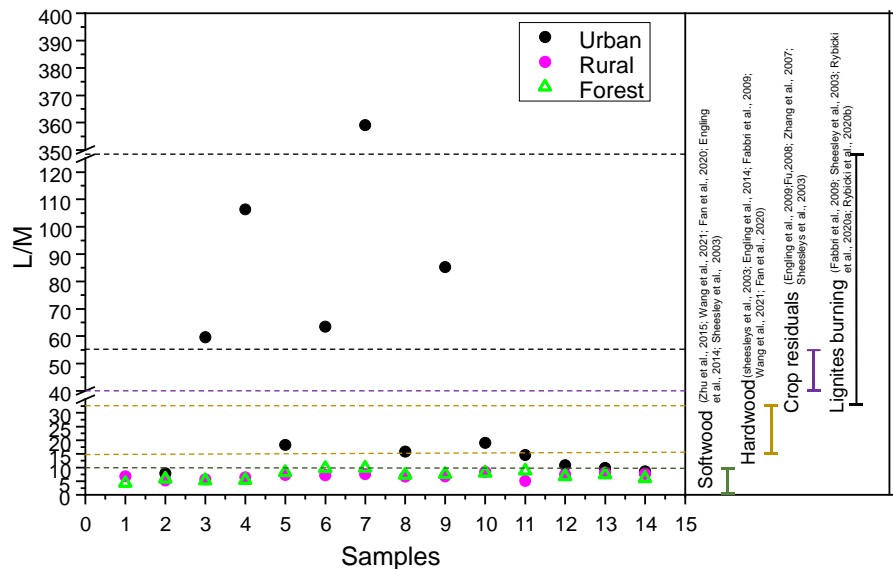

**Figure 10. Ratios of levoglucosan to mannosan in PM$_{2.5}$ and source test emissions from softwood, hardwood, crop residuals and lignites burning.**
