# Peer review of "Measurement Report: Characterization of sugars and amino acids in atmospheric fine particulates: identify local primary sources"

_Atmospheric Chemistry and Physics, 2022_

## Author Comment (AC1)

Dear Editors and Reviewers:

Thank you for your letter and for the reviewers' comments concerning our manuscript entitled "Measurement Report: concentrations and composition profiles of sugars and amino acids in atmospheric fine particulates: identify local primary sources characteristics" Those comments are all valuable and very helpful improving our paper. We would also thank the editor for providing an opportunity to revise of this manuscript. We have revised the manuscript according to the referee's points as follows.

Reviewer #1 Evaluations:

In this manuscript, sugars (anhydrosugars, primary sugars and sugar alcohols) and amino acids (free and combined) were determined in atmospheric fine aerosol particles (PM 2.5) collected from urban, rural and forest locations. The multiple trends observed for the different compounds and locations and their potential correlations were widely discussed. In addition, positive matrix factorization was used for the identification of local primary sources.

It is well known that organic aerosols (OA) have a clear impact on the climate. In this way, the determination of water-soluble compounds such as sugars and amino acids in OA collected from different geographical locations might be considered of great relevance to elucidate potential emission sources. Different statistical techniques such as Pearson correlation or positive matrix factorization were used to stablish the potential differences between the samples collected at different locations, identify the potential emission sources and quantify their contribution to the total aerosols. In general terms, results achieved in this research are promising and they could be of great interest for the atmospheric chemistry and physic researchers. However, there is some points related with the manuscript structure and the use of the different statistical tools that might be improved and/or clarified.

On one side materials and methods section includes a whole subsection devoted to sugar analysis. Almost no information (3 lines) has been included about the analysis of free and combined amino acids. It could be interesting provide a reliable summary of the analytical methods on the main text. Detailed information might be provided as reference or in the supplementary information section. On the other side, it could be difficult for the potential readers a detailed evaluation of the results showed in results and discussion section. Most of the results achieved for the analysis of the sugars on the samples has been included as supplementary information. However, the results achieved for amino acids has been included as figures in the main text. This could be a bit confusing even if some of the results obtained for the sugars are also shown in these figures.

Furthermore, it should be clarified if data distribution has been evaluated by the authors. Normal data distribution is required for most of the used algorithms and no-information is provided in the text.

Finally, the use of certain terms as "good correlation" or "highly correlated" is not clear and it should be reconsidered.

Therefore, this manuscript is not deserved for its publication in Atmospheric Chemistry and Physics in the present form, but it could be published after clarification of the following points:

**Answer:** Thank you for your suggestion. We have carefully revised our manuscript. A reliable summary of the analytical methods of FAAs and CAAs have been added in the supplementary information section. Moreover, combined with your subsequent suggestion about the number of figures being pretty large, two figures in the manuscript have been deleted. The number of figures in the revised manuscript is eight now.

The result of Normality test has been provided in the section of statistical analysis. "Normality tests of Shapiro–Wilk were performed. The concentrations sugar compounds (anhydrosugars, primary sugars and sugar alcohols), FAAs and CAAs are normally distributed."

Furthermore, the use of certain terms as "good correlation" or "highly correlated" were avoided in the revised manuscript.

Abstract

**This section is pretty large and it should be condensed. However, quantitative results are mostly missing in the abstract.**

Answer: Thank you for your suggestion. The abstract section has been condensed and important quantitative results are added.

**Line 18. Sampling locations should be specified in the text.**

Answer: Sorry for our unclear description. Sampling locations were specified in the revised manuscript.

"fine particulate matter samples were collected from the urban, rural and forest areas in Nanchang, China."

**Line 20. Analytical technique used for the determination of the compounds might be clarified in the text**

Answer: Sorry for our unclear description. Analytical technique used for the determination of the compounds has been defined in the text.

"The concentrations and compositions of sugar compounds (anhydrosugars, primary sugars and sugar alcohols), free amino acids (FAAs) and combined amino acids (CAAs)

were analyzed by gas chromatography–mass spectrometry (GC-MS) after silylation derivatization."

**Line 28. This abbreviation (L/M) should be defined in the text before it first use.**

Answer: Thank you for your suggestion. The abbreviation (L/M) has been defined in the text. Line 30-31.

"the average ratio between levoglucosan and mannosan (L/M)"

**Line 32. R2 and p-values should be included in this section for correlations.**

Answer: Sorry for our unclear description. Pearson Correlation Coefficient (r) and p-values have been added. Line 35.

"The concentrations of sugar alcohols in the rural and forest areas were positively correlated with that of CAAs, which are abundant in the topsoil ($r = 0.53 \sim 0.62$, $p < 0.05$)"

Introduction

**Lines 66-67. Not a clear link with the previous sentences.**

Answer: Thank you for your suggestion. This sentence was deleted in the revised manuscript.

**Line 96. This abbreviation (HAAs) should be defined in the text before it first use.**

Answer: Sorry for our unclear description. This abbreviation (HAAs) was defined in the text. Line 90.

"Combined compound-specific $\delta^{15}N$ patterns of hydrolyzed amino acids (HAAs)"

**Lines 104-109. This paragraph is a bit confusing; it could be interesting to clarify the aims of the study and then the different methodological improvements.**

Answer: Thank you for your suggestion. This paragraph was rewritten. The structure has been adjusted. The purpose of the research is first explained, and then the method improvement is further explained. Line 99-104.

"To further clarify local primary source information in different locations, the concentration and composition of sugars, FAAs and CAAs in $PM_{2.5}$ samples collected from the urban, rural and forest areas in Nanchang, China were analyzed. In order to obtain a sufficient sample size for analysis of sugar compounds (anhydrosugars, primary sugars and sugar alcohols), FAAs and CAAs in one sample simultaneously, in

this study different filter samples were collected during the same sampling period as our previous study (Zhu et al., 2020b)."

**Line 106. Reference should be included in the text.**

Answer: Thank you for your suggestion. The reference was added in the revised manuscript.

"FAAs and CAAs in one sample simultaneously, in this study different filter samples were collected during the same sampling period as our previous study (Zhu et al., 2020b)."

**Materials and methods**

**Line 121. Avoid the use of numbers at the beginning of the sentences.**

Answer: Special thanks for your careful checks. Sorry for our mistake. This sentence was rephrased.

"Forty-two samples were collected daily with a duration of 23.5 h on prebaked quartz fiber filters (450 °C, 10h)"

**Line 125. The time between the sampling and the transport to the laboratory should be clarified in the text. In addition, storage temperature in this step should be clarified in the text.**

Answer: Sorry for our unclear description. The time between the sampling and the transport to the laboratory and storage temperature in this step were defined in the revised manuscript.

"After the sampling the filter was recovered into a pre-combusted glass jar with a Teflon-lined screw cap, immediately placed in the freezer (the temperature is set to -20°C) and then transported to the laboratory within 1 hour and stored at -20°C before analysis, to prevent loss of semi-volatile/volatile organic compounds from the samples."

**Line 130. Have the authors considered the adsorption of the target analytes on the quartz wool during the sample preparation.**

Answer: Thank you for your suggestion. We have considered the adsorption of the target analytes on the quartz wool during the sample preparation. Recovery experiment of target analytes were performed. If large quantity of target analytes were adsorbed to the quartz wool, the recoveries of analytes will be very low. However, recoveries for sugar standards ranged from 89% to 108% (Table S2), which fall within the range of previous studies (Pietrogrande et al., 2013; Medeiros and Simoneit, 2007). Recoveries for AAs standards were provided in our previous study, which ranged from 83% to 102% (Zhu et al., 2020b).

**Line 149. At least R2>0.99 are required for quantitative purposes.**

Answer: Thank you for your carefully review. The linearity of sugar compounds is established simultaneously with the sample measurement. It may be more accurate to use the standard curve created with the samples than to recreate the standard curve now because the current instrument and environment conditions may be different from when the samples were measured. In order to solve this problem, we narrowed the linear range of some sugar compounds according to the actual concentration of the aerosol samples. For some sugar compound, their $R^2$ of the calibration curves is above 0.99 in a narrow concentration range.

"In general, the obtained calibration curves show good linearity ($R^2>0.99$) in the exploited range (Table S2)."

**Line 150. Preparation of field blanks should be clarified at some point (text or supplementary information)**

Answer: Sorry for our unclear description. Preparation of field blanks was added in the revised manuscript.

"The field blank samples were collected by using a modified version of (Chow et al., 2010). After treated at 450 °C for 10 hours, field blank filters were loaded into filter holders and accompany the sampled filters to each sampling site. Field blanks accompanied sample shipments and were placed in the high-volume air samplers along with the sampled filters. The passive period for field blank filters was consistent with the samples. The only difference between samples and field blanks is that air is not drawn through field blanks. After the sampling, field blank filters were also recovered into filter holders and accompanied with the samples back to the laboratory. Three field blank samples were collected in this study, accounting for 7% of total sample number."

"The field blank filters were also analyzed by the procedure mentioned above in this section and then were run in parallel with the samples for all analyses in order to monitor significant background interferences (Urban et al., 2012). No signal different than that of the base line was obtained for any of the studied sugars. The concentrations of sugar reported here are corrected for the field blanks."

**Line 155. Results should be avoided in the material and methods section. This information might be provided at results and discussion section.**

Answer: Thank you for your suggestion. We have deleted the results of recoveries, reproducibility and the detection limits of sugars in the material and methods section. Considering the length of the results and discussion sections of the article, these results have been described in the supplementary information. A new section of "Analytical characteristics for sugar compounds " was added in supplement.

**Line 166. Additional information about the selected methodology might be interesting for the potential readers.**

Answer: Thank you for your suggestion. Additional information about the methodology of the water-soluble ions concentration determination was added in the revised manuscript. Line 164-181.

"Water-soluble ions ($Na^+$, $NH_4^+$, $K^+$, $Mg^{2+}$, $Ca^{2+}$, $Cl^-$, $NO_3^-$ and $SO_4^{2-}$) were measured by ion chromatography (Dionex Aquion (AQ)™, ICS-90, Thermo Scientific™, USA)(Guo et al., 2020). Briefly, one eighth quartz filters were cut into pieces and transferred to a Nalgene tube. After added 50 mL Milli-Q water, the tube was ultrasonic vibration (30min), shaking (30min), and centrifugation (10min). Then, the supernatant was filtered by using pinhole filter with a 0.22μm microporous membrane. The extract was stored in a refrigerator at -20°C for analyses. The anions ($Cl^-$, $NO_3^-$ and $SO_4^{2-}$) were determined by the ion chromatography system with an AS23 4 × 250 mm analytical column, while the cations ($Na^+$, $NH_4^+$, $K^+$, $Mg^{2+}$ and $Ca^{2+}$) were determined using the same IC system with a CS12A 4 × 250 mm column. A sample volume was 100 μL for both anion and cation analyses. Strict quality control was performed to avoid any possible contamination of the samples. In order to ensure the reliability of data, standard samples were re-measured every twenty samples. The standard solutions were bought from Merck, Germany."

**Lines 167-169. Results should be avoided in the material and method section.**

Answer: Thank you for your suggestion. We move this section into the supplementary materials.

"Analytical characteristics for water-soluble ions
Standard solutions of water-soluble ions ($Na^+$, $NH_4^+$, $K^+$, $Mg^{2+}$, $Ca^{2+}$, $Cl^-$, $NO_3^-$ and $SO_4^{2-}$) were used for making the external standard curve before the analysis of $PM_{2.5}$ samples. The correlation coefficients of the calibration curves were greater than 0.999. The detection limit of $Na^+$, $NH_4^+$, $K^+$, $Mg^{2+}$, $Ca^{2+}$, $Cl^-$, $NO_3^-$ and $SO_4^{2-}$ were 0.001 μg $L^{-1}$, 1.21 μg $L^{-1}$, 1.77 μg $L^{-1}$, 2.47 μg $L^{-1}$, 0.09 μg $L^{-1}$, 5.1μg $L^{-1}$, 21.6 μg $L^{-1}$ and 11.5 μg $L^{-1}$, respectively. The relative standard deviation of the reproducibility test was less than 5% (Guo et al., 2020; Wen et al., 2022)."

**Results and discussion**

**Line 213. It could be interesting to clarify if reported results can be associated to pre- or post-pandemic samples.**

Answer: Thank you for your suggestion. It is very regrettable that the results reported here cannot be linked with the samples pre and post the epidemic. $PM_{2.5}$ samples in this study were collected at Nanchang, China from April 30, 2019 to May 13, 2019.

However, the first outbreak of COVID-19 in China occurred in Wuhan in December 2019. The sampling period in this study is 6 months before the outbreak of the COVID-19 pandemic, so it is very regrettable that our sampling period is not pre- or post- the epidemic.

**Lines 276 and 278. The selection of these groups of amino acids for study might be clarified in the text.**

Answer: Thank you for your suggestion. The selection of these groups of amino acids for study was clarified in the revised manuscript. Furthermore, the sequence of this section was adjusted.

"Furthermore, the temporal variations in the concentrations of source-specific sugars (including anhydrosugars, primary sugars and sugar alcohols) were compared with those of specific groupings of CAA species in $PM_{2.5}$. CAA species were initially grouped by their correlation with source-specific sugars at three sampling sites (Figure 6) and their abundance in potential emission sources (Figure S7)."

[Figure]

**Line 300. The use of the combination of Gly and Phe for the evaluation of the results should be clarified in the text.**

Answer: Thank you for your suggestion. The use of the combination of Gly and Phe for the evaluation of the results was clarified in the revised manuscript.

"To better understand the influence of primary sources on the composition profiles of sugar compounds and CAAs, the percentage of CAA species abundant in specific primary sources was compared with that of the corresponding sugar molecular markers. As discussed above, combined Gly and Phe are the predominant CAA species in the

aerosols from BB sources and they exhibited similar time variation patterns with molecular markers for BB sources (anhydrosugars). Therefore, their percentage were compared with the contribution of BB molecular marker to the total sugar compounds in three sampling sites. These two CAAs together accounted for an average of 19.7%, 23.9% and 20.4%, respectively, of the TCAAs pool in $PM_{2.5}$ collected from the urban, rural and forest areas. It is interesting to note that the percentage of Gly and Phe in total CAAs pool was generally similar to the percentage of the anhydrosugars in total sugar compounds in $PM_{2.5}$. "

**Line 338. Not clear that this is a good correlation. A more appropriate term might be correlation at 95%. In addition, the number of samples included in the model should be provided in the text.**

Answer: Sorry for our unclear description. This sentence was rewritten. Line 379-381.

"On the contrary, significant correlations at 95% confidence level ($r=0.56\sim0.57$, $p < 0.05$, $n=14$) were found between total combined amino acids (TCAA) and total sugars at each sampling site in this study (Figure 7a)."

**Lines 405-406. These reference should not be considered as recent references.**

Answer: Thank you for your suggestion. It has been changed to "Previous studies indicate that the amino acid Phe is not only used as vital constituent of proteins but also a critical metabolic node that plays an essential role in lignin biosynthesis in vascular plants (Bonawitz and Chapple, 2010; Wallace and Fry, 1994)." Line 444-445.

**Line 428. References from previous studies should be included in the text.**

Answer: Thank you for your suggestion. References from previous studies were added. Line 469-470.

"Consistent correlations were observed between sugar alcohols and combined Ala, Val and Leu at the rural and forest sites (Figure 7, green shadow), pointing to the significant contribution of biological sources associated with surface soil to aerosols at these two sites (Zhu et al., 2020; Moura et al., 2013; Fu et al., 2013; Simoneit et al., 2004)."

**Lines 493-494. Results obtained for L/M should be expressed homogeneously.**

Answer: Thank you for your suggestion. This section was rewritten.

"Levoglucosan/mannosan (L/M) ratios have been employed to identify specific burning substrates in numerous studies (Fabbri et al., 2009; Fan et al., 2020; Zhu et al., 2015).

L/M ratios for smoke particles from the burning of crop residues (~40 to 55.7) were high, while L/M ratios for the burning of hardwood (15-33) and softwood (3 to 10) were low (Engling et al., 2014; Engling et al., 2009; Sheesley et al., 2003; Wang et al., 2021). Besides that, recently studies demonstrate that levoglucosan is also present in lignite smoke particle matter (Rybicki et al., 2020a). Moreover, pyrolysis experiments confirm that lignite produce levoglucosan concentrations fully comparable to that of hardwoods and softwoods (Fabbri et al., 2009; Sheesley et al., 2003). Burning of lignite has been proved to be an additional input of levoglucosan to the atmosphere in regions where brown coal is utilized. Rybicki et al. (2020b) reported that the L/M ratios in lignites smoke (ranged from 31 to 189) were generally higher than those found for other fuel types because cellulose is more resistant to diagenetic (bio)degradation than hemicelluloses."

**Line 501. Please check the use of capital letters at the beginning of the sentences.**

Answer: Thank you for your suggestion. Based on the comments provided by reviewer 2, source apportionment based on PMF mode was deleted in this manuscript because the outcome of source apportionment may have large uncertainties. This sentence was also deleted with the deletion of PMF model results. So, this problem doesn't exist now.

**The number of figures is pretty large. Some figure should be summarized or moved into supplementary information.**

Answer: Thank you for your suggestion. As we mentioned above, since the source apportionment based on PMF mode was deleted in this manuscript, Figure 8 and 9 were also deleted along with the deletion of this section. However, more efforts were put on the identification of biomass burning source. A separate section "Identify classes of combustion sources" has been added in the revised manuscript.

---

## Author Comment (AC2)

Dear Editors and Reviewers:

Thank you for your letter and for the reviewers' comments concerning our manuscript entitled "Measurement Report: concentrations and composition profiles of sugars and amino acids in atmospheric fine particulates: identify local primary sources characteristics" Those comments are all valuable and very helpful improving our paper. We would also thank the editor for providing an opportunity to revise of this manuscript. We have revised the manuscript according to the referee's points as follows.

Reviewer 2

Zhu et al. investigated the concentrations and compositions profiles of sugars and amino acids in atmospheric fine particles to identify the source contributions at the urban, rural and forest sites in Nanchang, China. Sugars including anhydrosugars, primary sugars, alcohol-sugar were studied here as some tracers of sources. Together with the results of combined and free amino acids, the sources were identified using correlation analysis and a receptor model of PMF. It is a nice presentation on field measurement of organic tracers to suggest the possible local sources. My concerns are show as follows. I hope they could help improve the quality of this manuscript.

**I do not think that PMF is suitable for the source apportionment in this study. We should be aware that there are only 14 samples at every site. It may cause large uncertainties on the outcome of source apportionment. The lack of secondary reaction tracers would easily underestimate the contribution of secondary processes. The results in Figure 8 also clearly show that the source profiles are very different even they are attributed to a certain source. The results shown in Figure 9 may not be reliable due to the highly possible uncertainties. It is not recommended to keep PMF source apportionment in this manuscript.**

Answer: Thank you for your suggestion. The section of PMF source apportionment in this manuscript has been deleted and a new part of discussing biomass burning identification was added.

**I found that the temporal variations of the organic tracers are quite different at the three sites e.g. total sugars and amino acids in Figure 1. The variations have not been clearly presented and discussed in the text. The difference may point to the contribution of sources along the sampling period.**

Answer: Thank you for your suggestion. The difference in the temporal variations in the organic tracers among the three sites was further discussed in the revised manuscript. A new section of "Temporal variation of ambient sugars tracers and AAs" was added in the manuscript.

"The temporal variations of sugar and AA compounds in the PM$_{2.5}$ samples collected during the sampling periods are quite different at the three sites (Figure 1). In the urban

area, the total sugar concentrations showed two major peaks, one during 4-6 May and the other during 10-12 May. Anhydrosugars, the marker of BB source, showed similar trend with the highest concentration occurred during 4-6 May and 10-12 May, suggesting urban sites may be more influenced by BB sources. In the rural area, the temporal variations of the total sugar concentrations showed different patterns, with two major peaks on 3 May and 6 May. On these two days, sugar alcohols and primary sugars exhibited the highest concentration, indicating a significant contribution from primary biogenic sources in the rural location. In the forest area, total sugars showed its concentration maximum on 2 May and 5 May. The concentrations of sugar alcohols and primary sugars in the forest area showed similar trends to total sugars, with the highest concentration occurring on 2 May and 5 May, implying that this area may be greatly influenced by primary biological sources.

Figure S8 presents the overall temporal variations of three classes of sugar compounds (anhydrosugars, primary sugars and sugar alcohols) in $PM_{2.5}$ collected at the three sampling sites. The temporal patterns of biomass burning tracers (anhydrosugars), primary bioaerosols tracers (primary saccharides) and surface soil and associated biotas tracer (sugar alcohols) are similar in the forest area. In the rural area, the temporal variation of primary saccharides is similar to that of sugar alcohols, but it is not consistent with that of anhydrosugars. In contracts, anhydrosugars, primary sugars and sugar alcohols showed different trends in the urban area. This further supported that the difference in the temporal variations of sugar compounds in $PM_{2.5}$ may point to the contribution of sources along the sampling period. The sugar compounds in $PM_{2.5}$ collected in the urban area might be more influenced by combustion sources whereas ambient sugars in the forest area are more influenced by primary biogenic sources.

The covariation of TCAA concentration with total sugar concentration was found in the three sampling areas. On the contrary, the temporal variations of TFAA concentrations showed different patterns from total sugar concentrations. These suggest that CAA may have similar sources and atmospheric processes to sugar compounds in $PM_{2.5}$, whereas FAA may have different sources and atmospheric processes from them, which will be discussed further in the following section."

**Based the conclusions and discussions as well as the associated studies, biomass burning is expected and suggested as one of the major sources. I think there more efforts should be put on the identification of this specific source. A separate section is suggested. By the way, the discussion on sources of lignite combustion and road dust seem vague and should be improved.**

Answer: Thank you for your suggestion. A new section of "identify classes of combustion sources" has been added in the manuscript. The sources of lignite combustion have been discussed further in the revised manuscript. The discussion on sources of road dust in urban area was added in the section 3.1.2.

[revised manuscript text omitted]

**The authors claimed that some specific combined amino acids represent certain source contribution. For example, in Line 274, "anhydrosugars, combined Gly and Phe may be influenced by the identical source". It is really not the case because CAA(s) are hydrolyzed products from certain proteins or peptides. One or two CAA(s) may not be released from a certain sources.**

Answer: Sorry for our unclear description. There is an error in our description here. The concentration of some specific combined amino acids cannot represent certain source contribution. The revised manuscript grouping principle for CAAs was sufficiently described to clarify, and a graph for the grouping principle of CAAs was added (Figure S7). In this study, a hydrolysis method was used to measure CAAs in aerosol samples.

After complete hydrolysis, the peptide bonds in the protein are broken, so that the combined amino acids that constitute the protein are released as free amino acids. As your suggested, not only one or two specific CAA(s) are released from a certain source. However, previous studies found the protein CAA compositions differs in different primary biological emission sources. For example, Combine Gly, Ser, and Gln are demonstrated to be abundant CAA species in barley and cereals (Filippo et al., 2014). In our previous study, combined Gly was found to be the predominant CAA in the aerosol from biomass burning sources, hydrophobic species (Ala, Val, Leu, and Ile) were the major CAA species in local soil sources, and hydrophilic species (Glu, Lys, and Asp) were the abundant CAA species in local plant sources (Zhu et al., 2020b). (Abe et al., 2016; Miguel et al., 1999) suggested that the composition profiles of aerosol CAAs highly depended on the composition profiles of CAAs in major emission sources. If one primary biological emission source contributes significantly to the aerosols, the concentration and abundance of the CAA species predominant in this primary biological source will be higher than those of the other CAA species in $PM_{2.5}$. Furthermore, the CAA species predominant in this primary biological source may positively correlate with molecular marker of this emission source and exhibit a similar temporal variation pattern of the specific molecular marker. Therefore, the correlation and similar temporal variation pattern of the CAA species abundant in one primary biological emission source with the corresponding source specific molecular marker as well as the comparison of these CAA species with other CAA species in the aerosol may provide information on local primary biological emission source (Figure S7). The corresponding sections have been revised in the revised manuscript. Line 276-299.

[Figure]

**Line 243-247: I do not understand why? Trehalose is not a specific tracers of road dust.**

Answer: Sorry for our mistake. The impact of resuspension of surface soil is missing in the text. This sentence has been rewritten.

"Since trehalose is proved to be the most abundant saccharide in soils, the higher fraction of trehalose in total sugars pool in the urban location than those in the rural and forest locations may indicate $PM_{2.5}$ collected in the urban location may be more impacted by resuspension of surface soil or road dust outflow than the rural and forest locations (Simoneit et al., 2004a,b; Medeiros et al., 2006b; Rogge et al., 2007)(Fu et al., 2012). This is further supported by the significantly higher average concentration of nss-$Ca^{2+}$ observed in urban site (1834.3 ng m$^{-3}$) than that of rural (1027.8 ng m$^{-3}$) and forest (1130.7 ng m$^{-3}$). nss-$Ca^{2+}$ is reported to be mainly derived from resuspended dust/soil in the ambient air (Barbaro et al., 2019; Rathnayake et al., 2016). "

**Line 278: Why the sum of Ala, Val, Leu and Ile are calculated? As I mentioned above, they are dependent on the proteins and peptides in aerosol samples but we should have known nothing of them in this cause.**

Answer: Ala, Val, Leu and Ile are found to the major CAA species in local crop soil (Zhu et al., 2020). Thus, the sum of Ala, Val, Leu and Ile were calculated and compared with sugar alcohols (surface soil and associated biotas tracer).

**Line 284-289: Why they are separated into four groups? A reason is necessary.**

Answer: Thank you for your suggestion. CAAs are grouped by the positive correlation between source-specific sugar markers and specific CAA species, and the similar temporal variation patterns between source-specific sugars and specific CAA species, as well as the predominant specific CAAs species predominant in each local primary biological source, ambient CAAs were finally divided into 4 groups.

**The text in Line 312-325 presents some background information. It can be shortened.**

Answer: Thank you for your suggestion. This part has been shortened in the revised manuscript.

"This could be explained by the formation pathways and atmospheric processes of aerosol FAAs and sugars. The fact that secondary processes is an important formation pathway of FAAs in the atmosphere has been corroborated by many studies (Wang et al., 2019; Xu et al., 2020b). Both laboratory simulations and field measurements demonstrated that the oxidation of proteins and peptides with hydroxyl radicals (·OH) and O3 can release of FAAs (Liu et al., 2017b; Mcgregor and Anastasio, 2001; Song et al., 2017). On the other hand, sugar compounds are suggested as atmospheric primary organic aerosol, which directly emitted from sources such as plant material, soil dust,

microorganisms and BB, and then aerosolized (Fröhlich-Nowoisky et al., 2016; Joung et al., 2017)."

**Maybe the authors should rephrase the title. Make it clear and concise**

Answer: Thank you for your suggestion. The new title is "Characterization of sugars and amino acids in atmospheric fine particulates: identify local primary sources."

All changes were also highlighted in yellow color in the revised manuscript. Thank you very much again.

Yours sincerely,
Ren-Guo Zhu, Hua-Yun Xiao, Liqin Cheng, Huixiao Zhu, Hongwei Xiao, Yunyun Gong

---

## Author Response (AR2)

Dear Editors:

Thank you for your letter and comments concerning our manuscript entitled "**Measurement Report: Characterization of sugars and amino acids in atmospheric fine particulates and related to local primary sources**" Those comments are very helpful improving our paper. We have carefully revised the manuscript according to the points as follows.

Suggested changes:

Line 17 (abstract): Should be changed to "sugar compound and amino acid (AA) pools"

**Answer: Thank you for your suggestion. It has been corrected. "evaluate their contributions to the total sugar compound and amino acid (AA) pool in different regions"**

Line 28 (abstract): I would suggest changing the word order to be "These suggest that plant and soil sources, as well as biomass burning (BB), provide important contributions ..."

**Answer: Thank you for your suggestion. The word order has been changed as your suggestion. "These suggest that plant and soil sources, as well as biomass burning (BB), provide important contributions to aerosol sugars and CAAs in three areas."**

Line 54: Suggest changing to "Organic aerosols (OA) are an important topic of study, because ..."

**Answer: Thank you for your suggestion. It has been corrected as your suggestion.**

Line 87: Remove the word "obviously"

**Answer: Thank you for your suggestion. "obviously" has been deleted.**

Line 89: Change "Till now" to "Until the present, few studies ..."

**Answer: Thank you for your suggestion. It has been revised.**

Line 130: Your response to the reviewer related to this line was important. Please add a sentence or two here.

**Answer: Thank you for your suggestion. Two sentences have been added. "The sampling period in this study is 6 months before the outbreak of the COVID-19 pandemic. $PM_{2.5}$ samples collected in this study are not pre- or post-epidemic samples."**

Line 136: Does pre-combusted mean "pre-baked"? Pre-combusted would imply they were burned (like a combustion engine)

**Answer: Thank you for your suggestion. It has been changed to "After the sampling the filter was recovered into a pre-baked glass jar with a Teflon-lined screw cap"**

Line 140: The first sentence in this line is incomplete. Please correct.

**Answer: Sorry for our unclear description. The first sentence in this line was rewritten. "The field blank samples were collected using a modified method described by Chow et al. (2010)."**

Line 352: Please add a sentence along the lines of what you explained to the referee about the justification of the 4 groups.

**Answer: Thank you for your suggestion. A sentence was added to explain why AAs were divided into four groups. "Since the composition profiles of aerosol CAAs highly depended on the composition profiles of CAAs in major emission sources (Abe et al., 2016), the CAA species predominant in this primary biological source may positively correlate with molecular marker of this emission source and exhibit a similar temporal variation pattern of the specific molecular marker."**

Line 548: Please change to "Levoglucosan has been frequently used as a tracer for biomass burning ..."
**Answer: Thank you for your suggestion. It has been corrected.**

The new Sections 3.3 and 3.5 could use a bit more work to refine the grammatical language.
**Answer: Thank you for your suggestion. We have carefully revised grammatical mistakes in the new sections 3.3 and 3.5.**

The title is still a bit confusing. I would suggest changing to something like: "Characterization of sugars and amino acids in atmospheric fine particulates and related to local primary sources"
**Answer: Thank you for your suggestion. The title has been revised to "Measurement Report: Characterization of sugars and amino acids in atmospheric fine particulates and related to local primary sources"**

Yours sincerely,
Ren-Guo Zhu, Hua-Yun Xiao, Liqin Cheng, Huixiao Zhu, Hongwei Xiao, Yunyun Gong